

# Phytobiomes are compositionally nested from the ground up

Anthony S. Amend[1], Gerald M. Cobian[1], Aki J. Laruson[2], Kristina Remple[3], Sarah J. Tucker[4,5], Kirsten E. Poff[6], Carmen Antaky[7], Andre Boraks[1], Casey A. Jones[1], Donna Kuehu[8], Becca R. Lensing[4,9], Mersedeh Pejhanmehr[1], Daniel T. Richardson[7] and Paul P. Riley[7]

[1] Department of Botany, University of Hawaii at Manoa, Honolulu, HI, United States of America
[2] Department of Biology, University of Hawaii at Manoa, Honolulu, HI, United States of America
[3] Department of Oceanography, University of Hawaii at Manoa, Honolulu, HI, United States of America
[4] Marine Biology Program, University of Hawaii at Manoa, Honolulu, HI, United States of America
[5] Hawaii Institute of Marine Biology, University of Hawaii at Manoa, Honolulu, HI, United States of America
[6] Plant and Environmental Protection Services, University of Hawaii at Manoa, Honolulu, HI, United States of America
[7] Department of Natural Resources and Environmental Management, University of Hawaii at Manoa, Honolulu, HI, United States of America
[8] Department of Molecular Biosciences and Bioengineering, University of Hawaii at Manoa, Honolulu, HI, United States of America
[9] Department of Microbiology, University of Hawaii at Manoa, Honolulu, HI, United States of America

Corresponding author
Anthony S. Amend,
amend@hawaii.edu

## ABSTRACT

Plant-associated microbes are critical players in host health, fitness and productivity. Despite microbes' importance in plants, seeds are mostly sterile, and most plant microbes are recruited from an environmental pool. Surprisingly little is known about the processes that govern how environmental microbes assemble on plants in nature. In this study we examine how bacteria are distributed across plant parts, and how these distributions interact with spatial gradients. We sequenced amplicons of bacteria from the surfaces of six plant parts and adjacent soil of *Scaevola taccada*, a common beach shrub, along a 60 km transect spanning O'ahu island's windward coast, as well as within a single intensively-sampled site. Bacteria are more strongly partitioned by plant part as compared with location. Within *S. taccada* plants, microbial communities are highly nested: soil and rhizosphere communities contain much of the diversity found elsewhere, whereas reproductive parts fall at the bottom of the nestedness hierarchy. Nestedness patterns suggest either that microbes follow a source/sink gradient from the ground up, or else that assembly processes correlate with other traits, such as tissue persistence, that are vertically stratified. Our work shines light on the origins and determinants of plant-associated microbes across plant and landscape scales.

## INTRODUCTION

Many of what we formerly considered ''plant'' traits we now know to be the direct or indirect result of a consortium of microbial species that colonize the inside and outside of plant tissues. Known as the phytobiome, these microscopic organisms from

throughout the tree of life play critical roles in plant chemistry, health, fitness and phenology (*Glick, 2014*; *Bulgarelli et al., 2015*; *Panke-Buisse et al., 2015*; *Hacquard et al., 2015*). Given microbes' central role in plant health, it is remarkable that most microbes are not inherited by birthright (like chloroplasts or mitochondria). Instead, most hosts emerge physically decoupled from their microbiomes, which are accumulated from the surrounding environment (*Hodgson et al., 2014*). This process may convey some advantages, as it enables a population to adapt to a local habitat more quickly than would be possible relying on the comparatively slow process of evolution (*Lau & Lennon, 2012*). For this reason, a plant's location can be a strong determinant of phytobiome composition (*O'Rorke et al., 2015*). However, microbial composition can be conserved *within* plant parts across multiple host species and environments (*Lundberg et al., 2012*; *Bodenhausen, Horton & Bergelson, 2013*; *Ottesen et al., 2013*; *Lambais, Lucheta & Crowley, 2014*; *Leff et al., 2015*; *Laforest-Lapointe, Messier & Kembel, 2016*; *Müller et al., 2016*). Two leaves, though oceans apart, might nevertheless recruit a similar consortia of microbes.

While researchers have expended considerable effort to study the assembly dynamics driving phytobiome structure and diversity, we have yet to uncover from where these microbes derive and how the relative contribution of spatial factors and plant part governs the microbial assembly on plants. Here, we examined how two factors: plant part identity and geographic distance, partition surface microbial communities. We chose these variables because they represent strongly deterministic and stochastic processes respectively, and because they are reasonably easy to isolate on a volcanic island when aspect and elevation are held constant.

Factors contributing to differences among plant parts are highly deterministic and form microhabitats that vary considerably at small spatial scales. Rhizospheres, for example, combine physiochemical properties of soils with nutrient and moisture inputs from plants (*Fitzpatrick et al., 2018*), whereas leaf surfaces can be hydroscopic and nutrient poor (*Remus-Emsermann & Schlechter, 2018*). Nectar producing flowers, on the other hand, often provide a sugar-rich, low water potential, acidic habitat (*Aleklett, Hart & Shade, 2014*). Tissue longevity might further differentiate plant microbes. Whereas stems may persist for the entire life of a plant, most other tissues have more limited lifespans. In some species (i.e., *Hibiscus trionum*, the so-called flower-of-an-hour), flowers are open for less than a single day, a seemingly short timeframe in which to recruit and establish a resident microbiome. Distance, in contrast, serves as proxy for dispersal limitation (*Peay et al., 2012*) or priority affects (*Kennedy, Peay & Bruns, 2009*; *Cadotte, Cardinale & Oakley, 2012*), which are stochastic environmental drivers assuming that other environmental conditions are held constant.

Further, we sought to determine whether within-plant distributions of microbes are consistent with microbial source sink dynamics. We hypothesized that soil and rhizosphere microbial communities serve as a reservoir for above-ground plant parts. By conceptualizing plant parts as nodes in a bipartite network we can determine whether microbial communities in ephermeral and late-emerging plant parts are compositionally nested within early-developing and long-lived plant parts, consistent with the hypothesis that a plant effectively inoculates itself from the ground up.

To address these questions, we examined how distance and plant part identity structure the surface bacteria of beach Naupaka, *Scaevola taccada*, a widespread littoral shrub native to Hawai'i and much of the tropics. Sampling at three scales (within plant, within site, and across island) we examine how distance and plant part shape phytobiomes, and make comparisons with adjacent soil communities that assemble independent of a biological host. Using a multivariate modeling framework, we determine the extent to which community variance is partitioned by site and by plant part. Finally, to gain insight into phytobiome source sink dynamics, we determine whether microbial communities are compositionally nested with respect to plant part.

We find that communities appear to be more sensitive to within-plant location, than to site differences. We find, unsurprisingly, a split between below-ground and above-ground samples, and distinct community structure on vegetative and reproductive plant parts, suggesting that plant part function and longevity may contribute to these differences. Phytobiomes within plants are compositionally nested, and demonstrate hierarchical structuring consistent with expectations of soils as plant microbial reservoirs.

## MATERIALS & METHODS

### Sampling

We selected our plant species and sites to minimize environmental heterogeneity. *S. taccada* is an evergreen shrub, native to Hawai'i, which produces fruits and flowers across seasons. It is also one of the most common littoral plants on Oahu's East (windward) shore, enabling sampling at regular spatial intervals. The coastal habitat was selected because local climates at this elevation are minimally influenced by land topography, and because all coastlines are accessible to the public by virtue of state law. All samples were collected on January 26, 2017. A single *S. taccada* individual was sampled from each of 10 locations along the windward shore of O'ahu at 6 km intervals (Fig. 1). Generally, there were few individuals containing both reproductive parts (fruits and flowers) at a given site, and if more than one was present the sampled individual was selected haphazardly. Within the Kailua site, nine additional individuals were selected haphazardly within a 50 m$^2$ plot. Plant locations were recorded with a Garmin Rhino GPS. A single mature, disease-symptom free leaf, flower, fruit, stem and axil was sampled by swabbing with a sterile cotton swab that had been moistened with an alkaline Tris extraction buffer (containing EDTA and KCL; comparable to Extract-N-Amp solution; Sigma Aldrich, St. Louis, MO, USA). At two locations we were unable to locate and sample flowers. Root (rhizosphere) samples were collected by selecting an area of the root with diameter 0.5–1.7 cm, and buried 2–10 cm below the surface. Soil was collected on swabs taken within 10 cm of the plant base, at a depth of 2–5 cm. Because soil was likely the most heterogeneous component of our study, bulk soil samples were also collected for soil description; including soil taxonomy and pH. Finally, sterile swabs dipped in extraction solution were exposed to the air for approximately 20 s as extraction negative controls. After sampling, swabs were immersed in 200 µL extraction buffer and stored in a cooler until DNA extraction later in the day. Although soil is not technically a plant part, we refer to it as such when differentiating sample types below.

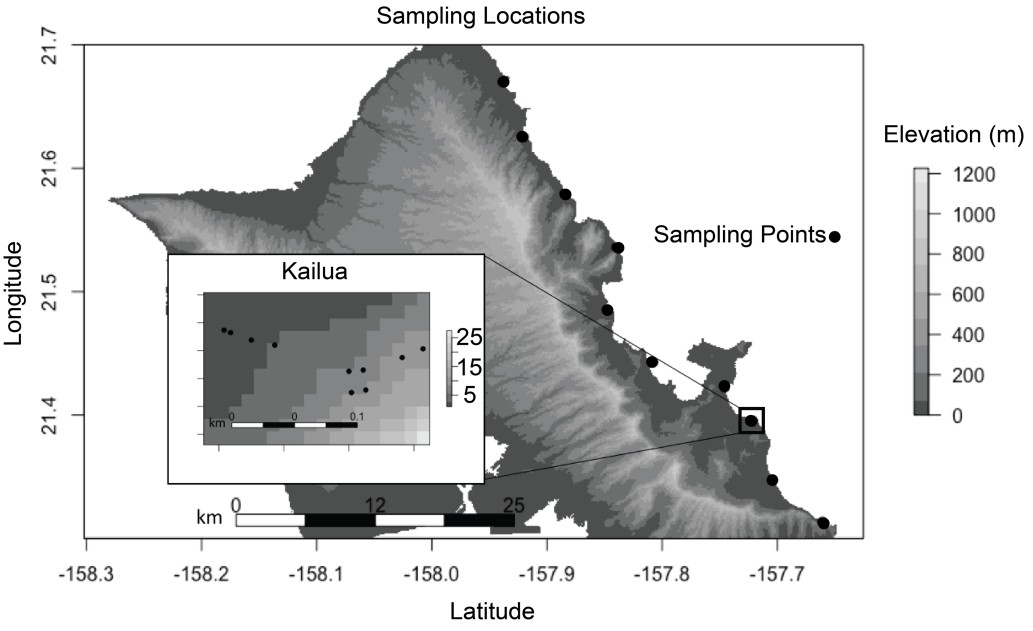

**Figure 1** **Each sampling location indicates where seven biological samples were collected (six plant parts and a soil sample).** Sites were spaced *ca.* 6 km along the windward coast. Ten sites were sampled in Kailua (inset). Axes represent decimal degrees. Topography is show to indicate that all samples were collected at sea level and with a similar aspect.

## Nucleic acid extraction and library preparation

Swabs were incubated in 50 µL of the extraction buffer at 65 °C for 10 min and at 95 °C for an additional 10 min. After brief centrifugation, swabs were removed from the buffer, and 200 µL of 3% BSA was added to each sample. DNA extractions were purified and concentrated using a carboxylated paramagnetic bead solution. Briefly, 100 µL of gDNA was mixed with 80 µL of bead solution, immobilized with magnets, washed twice with 70% ethanol, and suspended in 30 µL of 0.1x TE buffer.

## DNA library preparation and sequencing

PCRs targeted the v4 region of the 16S rRNA gene using the primers 515f (GTGCCAGCMGCCGCGGTAA) and 806r (GGACTACHVGGGTWTCTAAT) with overhangs complementary to the 3′ region of a construct containing dual 8bp indices and the Illumina i5 and i7 adaptors following the standard Illumina 16S Metagenomic Sequencing Library Preparation protocol (https://support.illumina.com/content/dam/illumina-support/documents/documentation/chemistry_documentation/16s/16s-metagenomic-library-prep-guide-15044223-b.pdf; Illumina, San Diego, CA), using KAPA3G plant PCR kit reagents. Briefly the first PCR was carried out in a 25 µL reaction consisting of 25 cycles. Amplicons were diluted 1:10 and used as a template for the second "Index PCR" in which sample-specific illumina indices and sequencing adapters were added using an additional 8 cycles.

Negative PCR controls (extraction and field blanks), were added to the library preparations. PCR products were purified and normalized to equimolar concentrations using the just-a-plate 96 PCR Purification and Normalization Kit (Charm Biotech, San Diego, California, USA). Normalized PCR products were pooled and sequenced at the Hawai'i Institute of Marine Biology CORE Laboratory using the Illumina MiSeq V3, 600 cycle, paired-end protocol. PhiX (15%) was added to the sequencing reaction to increase template complexity. A region of the flow-cell was deemed defective after its use, resulting in fewer high-confidence sequences data than are typically recovered using this technology. Sequence data were deposited to the SRA as PRJNA385181.

## Data processing

Average reverse sequence quality scores declined to <25 after *ca* 150 bp and R2 sequences were, therefore, not further considered. Chimeras were detected and removed from 16S sequences using VSEARCH (*Rognes et al., 2016*). Sequences were filtered by length (75 bp min) and quality score (mean score 25) and demultiplexed within the QIIME environment (*Caporaso et al., 2010*). OTU binning at 97% identity and taxonomic assignments were conducted in QIIME's "open reference" workflow using default parameters (*Caporaso et al., 2010*) and the Greengenes 13_8 reference database. Taxa that were not assigned to bacteria (chloroplasts, and two mitochondria sequences) were removed from datasets, as were OTUs represented by ten or fewer sequences. None of the extraction or field negatives produced sequence data.

Because of uneven sequencing depth across our dataset, data were down-sampled to 1,000 reads per sample, a cutoff that included the majority of samples. Although there is no consensus about how best to treat uneven sampling depth, down-sampling to a common depth is generally robust for multivariate comparisons of community compositions (*Weiss et al., 2017*). Eighty-five samples containing greater than 1,000 reads were retained for subsequent analysis.

## Data analyses

To determine which factors (site and/or plant part) predicted community composition, binary Jaccard distances were used in a single PERMANOVA model (function "adonis" in the vegan package (*Oksanen et al., 2013*)) considering marginal values of both factors and their interactions. In order to balance the number of samples among sites, a single individual plant was selected for the Kailua site. This analysis was repeated, including all samples from the Kailua site with qualitatively similar results. Because the analysis can be sensitive to dispersion of beta-diversity values, these were evaluated separately for plant parts. Following a globally significant RDA analysis showing that community dissimilarity was predicted by site, plant part and their interaction, a variance partitioning approach was used to determine the proportion of variance explained by each.

To address whether soil or plant parts might serve as a microbial reservoir for self-inoculation, we used two measures of nestedness, both of which indicate the extent to which OTUs present in species-poorer plant parts are contained in those with higher species richness. A bipartite matrix was calculated by summing all reads within each plant part and

then downsampling to the lowest sample sum. We calculate the nestedness temperature (*Atmar & Patterson, 1993*), a popular, though older, method calculating nestedness as a range from 0 (perfectly nested) to 100 (perfectly random) based on hierarchical entropy order and presence absence data. We also calculated nestedness based on overlap and decreasing fill (NODF; *Almeida-Neto et al., 2008*) scaled from 0 to 1. Both were compared with a distribution of randomized null communities generated using the *quasiswap* method in vegan. Differences in community membership among habitats were decomposed into nestedness and turnover components (*Baselga & Orme, 2012*) as implemented in the *nestedbetajac* function in vegan.

The extent to which certain plant parts "select for" microbes, was calculated using the network-wide H2 index (*Blüthgen, Menzel & Blüthgen, 2006*), which returns a value from 0 (generalized) to 1 (specialized) based on potential associations given OTU abundance totals. Significant deviations from null expectations were quantified using a distribution of null community matrixes calculated using the quantitative *r2dtable* method in vegan in which marginal totals are maintained. We calculated the related d' statistic (*Blüthgen, Menzel & Blüthgen, 2006*) to assess selectivity of individual plant parts.

To test for dissimilarity by distance patterns, binary Jaccard community dissimilarities were compared with geographic distances between samples using Mantel correlations (tests using Bray Curtis indices were qualitatively identical). So that the intensively sampled Kailua site did not skew detected spatial structure, we randomly selected a single Kailua individual for analysis (Mantel correlations including all samples showed similar effect sizes, except for above ground plant parts, which no longer showed a significant pattern; Table S2).

Indicator species analysis was calculated using the *indicspecies* package (*De Caceres & Jansen, 2016*) on single groups (multi-group combinations were suppressed due to the high number of OTUs).

## RESULTS

### Sequencing results

Illumina sequencing of the V4 region of bacteria 16S gene generated 15,354,523 reads. Following demultimplexing, quality score filtering and chimera detection 8,901,385 bacteria reads remained. Removal of OTUs identified as chloroplasts and mitochondria, and OTUs present in read abundances <10 resulted in 4,643,809 reads and 2,192 bacteria OTUs retained for subsequent analysis.

### Specialization and nestedness of habitats

Bacteria were significantly more nested than null expectations (Table 1; Fig. 1) and nestedness contributed 9% of total between-sample dissimilarity. Nestedness hierarchy followed the vertical structure of the plant: below-ground samples contained the highest species richness, above-ground vegetative structures were intermediate, and reproductive structures were lowest in the hierarchy (Fig. 2).

Microbes were significantly specialized on particular plant parts. We examine the degree and nature of specialization across plant parts using a bipartite network architecture index, and also within plant part using an indicator species analysis. Network-wide degrees of

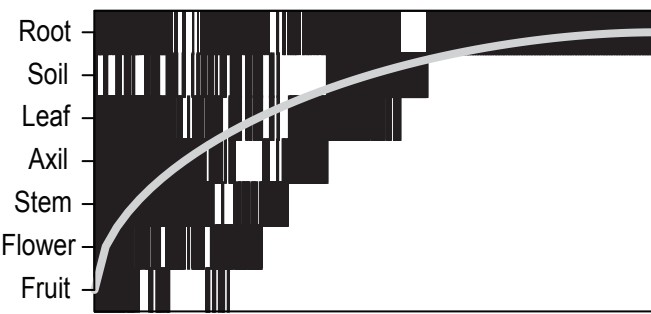

**Figure 2  Nestedness plot of bacteria aggregated by plant part.** Presence of an OTU in a plant part is represented as a rectangle. OTUs are ordered by occupancy (left to right) across plant parts, and rows are ordered by highest OTU richness (top to bottom). If all OTUs occurred above the "Fill line" (curved line), the network would be perfectly nested.

**Table 1  Network stucture of microbial communities.** The $d'$ statistic is a measure of specialization by habitat ranging from 0 (not specialized) to 1 (completely specialized). H2 is an index of specialization across all taxa within the network and is measured on the same scale. Both networks were significantly more specialized than randomized null simulations would predict. Plant parts are ordered by nestedness structure, with the most species-rich communities on the top. Turnover and nestedness proportions describe community dissimilarity among plant parts explained by that process.

| Sample | $d'$ | $P$ |
|---|---|---|
| Rhizosphere | 0.47 | 0.001 |
| Soil | 0.55 | 0.001 |
| Leaf | 0.29 | 0.001 |
| Axil | 0.31 | 0.001 |
| Stem | 0.34 | 0.001 |
| Flower | 0.25 | 0.001 |
| Fruit | 0.30 | 0.001 |
| Nestedness Temp. | 28.21 | 0.001 |
| NODF | 0.24 | 0.001 |
| H2 | 0.64 | 0.001 |
| Turnover | 0.84 | |
| Nestedness | 0.09 | |

specialization were high and 762 "indicator species" were statistically associated with plant parts (Table S3). The d' statistic, which measures degree of specialization for individual habitats, showed that soil and rhizosphere were the most specialized parts.

## Plant part and abiotic drivers of community composition

A PERMANOVA model demonstrated that both plant part and location shape *S. taccada* phytobiomes (Table 2), but that there was no interaction between those terms. Further examination of community partitioning demonstrates that plant part is a stronger determinant of bacteria community than location. Partitioning of variance, based on RDA analysis, demonstrated differences in the relative importance contributed by these two variables. Plant part accounted for 88% of the explained variance (RDA, $DF = 50$,

**Table 2  PERMANOVA examining community compositional variance explained by plant part, site, or their interaction.**

| Component | Degrees freedom | Sum of squares | F | P |
|---|---|---|---|---|
| Part | 6 | 5.1887 | 5.8391 | **0.001** |
| Site | 8 | 2.894 | 2.4426 | **0.001** |
| Part:Site | 29 | 6.6346 | 1.5447 | **0.012** |
| Residual | 4 | 0.5924 | | |

$F = 1.33$, $P = 0.043$), although residual variance accounted for 81% of the total, suggesting that other unmeasured variables may be more predictive of community composition.

NMDS ordination (Fig. 3) demonstrates that bacteria communities assorted into three principle groups associated with plant parts: belowground (soil and rhizosphere), mid-plant vegetative (stem and axil), and reproductive (fruit and flower). Leaf samples are spread among the other above-ground plant parts. Dispersion of bacteria beta diversity (bacteria betadispersion ANOVA, $DF = 6$, $P = 0.08$; Fig. S1) did not differ significantly between any sample types, suggesting that results of the PERMANOVA analysis reflect differences in multidimensional community "location" rather than spread. Bacteria were strongly partitioned at the order level by plant parts (Fig. 4). Bacillales were ubiquitous and abundant in soil and leaf samples, whereas Rhizobiales and Actinomycetales were abundant across root and vegetative plant parts.

While site location was a strong predictor of compositional differences, tests of dissimilarity by distance for microbial community compositions showed mixed results. There was a strong distance-decay relationship for below-ground samples and the relationship for above-ground plant parts and all samples was weaker, though significant (Table 3). Although geographic distance was generally a poor predictor of microbial composition, site location strongly impacted community composition (Table 2). This may be due to a deterministic influence of environmental variables that were not accounted for in our study, including the influence of the surrounding vegetation.

# DISCUSSION

## Distribution of OTUs across the plant

We show that the surface phytobiome composition of *S. taccada* is strongly shaped by plant part. Both community similarity and richness patterns are structured by plant parts: stem, axil, and reproductive parts were most similar to each other, rhizosphere samples were most similar to soil samples, and leaves were allied with above-ground parts more generally. Segregation of microbial communities by plant part (*Junker et al., 2011*; *Ottesen et al., 2013*; *Junker & Keller, 2015*) and even within-individual part location (*Leff et al., 2015*) has been reported for bacteria previously. Notably, *Leff et al. (2016)* demonstrated that both fungi and bacteria are partitioned by plant part (seed, rhizosphere and root) among sunflower cultivars. By examining differences among plant parts within a network context, however, we gain insight into the potential sources and sinks of plant microbes.

Compositional differences between samples might be attributed to both species replacement and nestedness components, and the relative contributions of these variables

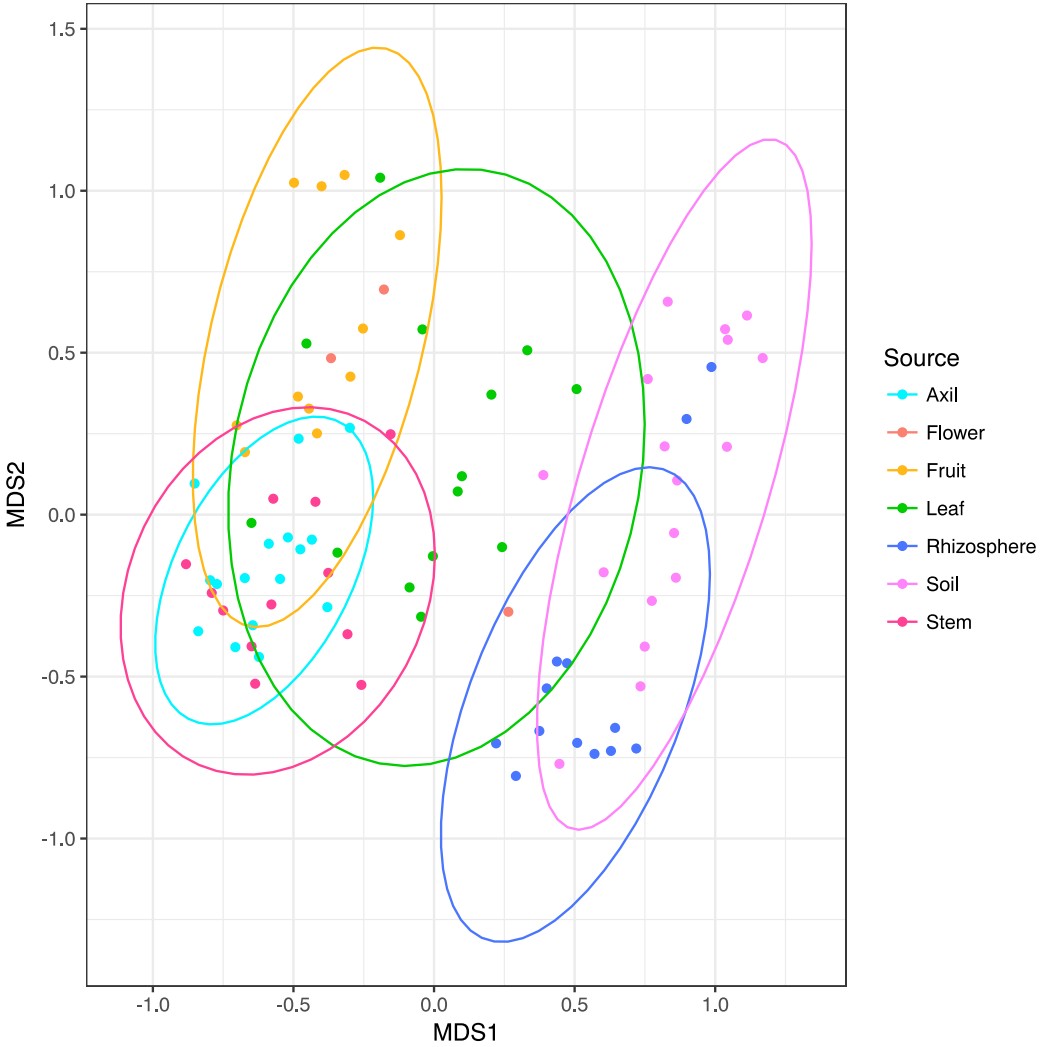

**Figure 3** **Non-metric multidimensional scaling plots of microbial communities colored by habitat.** Ellipses indicate 95% confidence intervals. Above and below-ground parts are differentiated along the first axis, with leaves intermediate.

can suggest causal mechanisms (*Baselga, 2010*). High turnover typically suggests stochastic assembly processes, whereas nestedness is more characteristic of deterministic factors such as environmental filtering (*Si et al., 2016*). We found that 9% of compositional differences was due to nestedness, a number that is consistent with the global mean of $0.093 \pm 0.054$ (SD), calculated in a recent meta-analysis of 99 studies, including microbes (*Soininen, Heino & Wang, 2018*). In contrast, the proportion of beta-diversity explained by turnover, 84%, was significantly higher than the global mean of $53\% \pm 18.4$ (SD).

We show that soil may be an important source for plant-surface microbiomes, including above-ground parts. Our analysis demonstrates that plant microbes are nested with respect to plant parts, that nestedness accounts for a large proportion of the among-sample diversity, and that below-ground parts (soil and rhizosphere) are at the top of the hierarchy.
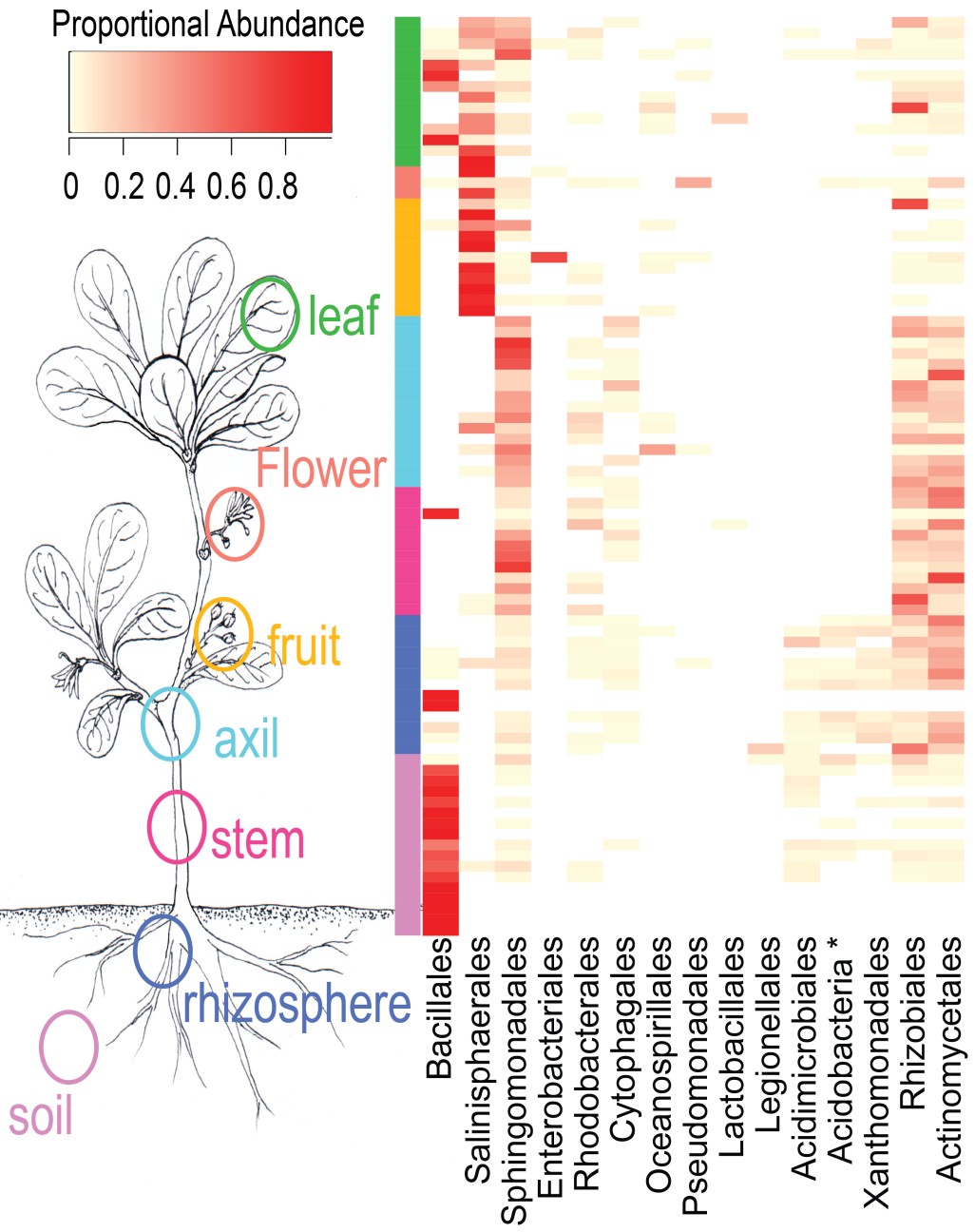

**Figure 4** **Heatmap of order-level taxa as distributed across plant parts.** Cell values are calculated proportionally across samples (rows).

Even putative "soil" dwelling microbes, such as Bacillales, are found throughout the plant. These results suggest that soils may serve as an important source for plant-surface microbes, either through initial inoculation as the plant emerges from the ground, or repeated introduction via wind and water dispersal. An alternative hypothesis, that soils and plants could both be sinks from an alternative source (such as airborne bacteria), with

**Table 3  Mantel testsmeasure the correlation between geographic distance and microbial community dissimilarity for *Scaevola taccada* surface microbes.** Sample sites occur as 10 locations spanning 60 km on the island of Oahu, HI. Mantel's statistic measure Pearson's product-moment correlation with 999 permutations. Significant values indicating a relationship between spatial distance and community dissimilarity are bolded. In this analysis a single plant individual was selected from the Kailua site. Above-ground refers to all plant parts, whereas belowground refers to rhizospere and soil.

| Plant part | *r*-value | *p*-value | *n* |
|---|---|---|---|
| All plant parts | 0.06538 | **0.046** | 48 |
| Above-ground | 0.1319 | **0.017** | 32 |
| Below-ground | 0.3781 | **0.008** | 16 |
| Soil | 0.1334 | 0.231 | 9 |

nestedness patterns resulting from ordered extinctions, could be tested via experimental manipulations.

Several other studies have noted nestedness patterns in other microbial systems, including mycorrhizal tree islands displaying strong species area relationships (*Peay et al., 2007*), and mouse guts whose bacterial diversity decreases, orderly, over generations (*Sonnenburg et al., 2016*). Notably, bacteria phyla from 97 independent studies spanning diverse geography and habitat were significantly nested (nestedness decreased with taxonomic rank; *Thompson et al., 2017*). Although this study did not consider plant parts *per se*, they also found significant nestedness patterning among plant samples, with rhizosphere being the most species rich category, followed by plant surfaces, and internal plant tissues.

Despite nestedness patterns overall, each plant part contained a distinct subset of indicator bacteria that were not shared. These patterns suggest that distinct assembly processes may govern different plant parts. Pollinators, for example, are known to be effective vectors for microbes found in nectar (*Herrera et al., 2010*; *Aleklett, Hart & Shade, 2014*). Less is known about the origins or mechanisms of inoculation of other plant parts.

Factors other than physiochemical plant traits also likely play a role in shaping phytobiomes. The strong division between below- and above-ground communities, for example, may result from dispersal limitation in addition to climatic differences. Furthermore, plant part longevity may play a role in compositional differences. Flowers and fruits, the plant parts with lowest species richness, are also the most ephemeral, present on a plant for days to weeks. *S. taccada* flowers studied on the French Polynesian Island of Moorea, for example, remain open for an average of fewer than four days (*Liao, 2008*), seemingly scant time to recruit microbes from the environment.

## Plant microbial distributions over space

In microbial biogeography studies, distance between samples is generally a reliable predictor of microbial community dissimilarity, particularly among communities that are not attached to a macro-organism (*Martiny et al., 2011*; *Zinger, Boetius & Ramette, 2014*; *Tedersoo et al., 2014*), even at the scale of centimeters (*Bell, 2010*). The distance decay of plant-associated microbial similarity is less resolved compared with studies of "host-independent" substrates such as ocean, water, or soil. A positive pattern could be attributed

to at least two factors and or their cumulative effects. First, dispersal limitation could lead to clinal dissimilarity among communities by enabling "ecological drift" over relatively short timescales (*Martiny et al., 2006*). Second, geographic distance could correlate with some environmental cline, which selects for a distinct adapted community. Factors such as rainfall (*O'Rorke et al., 2017*), soil pH (*Fierer & Jackson, 2006*), soil nutrients, or even host genotypic clines (*Bálint et al., 2013*) all contribute to community dissimilarity.

Among plant-associated microbes, evidence for the dissimilarity by distance pattern is mixed. *Redford et al. (2010)* found little evidence of geographic structure in pine bacteria phyllosphere communities spanning continents, and *Meaden, Metcalf & Koskella (2016)* found no evidence for dispersal limitation among Oak phyllosphere bacteria within a 20 ha plot. In contrast, *Stone, Bacon & White (2000)* found strong isolation by distance patterns among Magnolia phyllosphere bacteria located within 400 m of each other. *Oono, Rasmussen & Lefèvre (2017)* found that low abundance pine needle fungal endophytes were spatially structured over ∼100 km, but that high abundance communities were not.

In our study, both above-ground and belowground bacteria were correlated with geographic distance (Table 3), although the latter relationship was stronger. Differing community turnover rates among plant parts could be attributable to different dispersal rates since airborne bacteria would presumably travel farther and more quickly due to prevalent trade winds compared to subsurface soil bound communities. Although there were no obvious environmental clines along our transect, microbes may have responded differently to additional unmeasured variables. Multiple studies show that bacteria are particularly sensitive to soil properties such as pH. Substrates change rapidly over short distances from the high tide, and although pH did not covary in a linear fashion along the transect, pH was variable among sample sites. For example at a single site for this study, we sampled soils ranging from ph 5.28 to 8.65 (Table S1). Finally, differences in the distance decay relationship could be driven by interactions with the host plant. Obligacy or strong host selection of some community members, for example, could depress turnover if the microbe was required for growth and survival.

## CONCLUSIONS

Plant-associated microbes are critical players in plant fitness and health. Our work shows that while both stochastic and deterministic factors play a role in shaping surface phytobiomes, plant part is the most predictive, with nestedness patterns relating strongly to vertical stratification. Bacteria are highly differentiated across a plant, with distance playing a weak, though significant, role in community composition at the scale examined. Despite these patterns, we were unable to describe the majority of community variance in this system. Identifying environmental reservoirs for phytobiomes, particularly those sources for ephemeral reproductive parts, will help in understanding plant associated microbial distributions and the factors that shape them.

## ACKNOWLEDGEMENTS

The authors wish to thank Michele Langner of Illumina for donating sequencing reagents. We also appreciate the assistance and support of Amy Eggers and the Hawai'i Institute of Marine Biology Genetics Core Facility. We also appreciate the feedback of the handling editor and two anonymous reviews. This paper was the result of a class project: High Throughput Sequencing Approaches to Ecology and Evolution at the University of Hawaii at Manoa.

### Funding

This work was supported by National Science Foundation grant #1255972 to Anthony S. Amend and NSF fellowship #1329626 to Gerald M. Cobian. The funders had no role in study design, data collection and analysis, decision to publish, or preparation of the manuscript.

### Grant Disclosures

The following grant information was disclosed by the authors:
National Science Foundation: #1255972, #1329626.

### Competing Interests

Anthony S. Amend is an Academic Editor for PeerJ.

### Author Contributions

- Anthony S. Amend conceived and designed the experiments, performed the experiments, analyzed the data, contributed reagents/materials/analysis tools, prepared figures and/or tables, authored or reviewed drafts of the paper, approved the final draft.
- Gerald M. Cobian conceived and designed the experiments, performed the experiments, analyzed the data, prepared figures and/or tables, authored or reviewed drafts of the paper, approved the final draft.
- Aki J. Laruson, Kristina Remple, Sarah J. Tucker, Kirsten E. Poff, Carmen Antaky, Andre Boraks, Casey A. Jones, Donna Kuehu, Becca R. Lensing, Mersedeh Pejhanmehr, Daniel T. Richardson and Paul P. Riley performed the experiments, analyzed the data, prepared figures and/or tables, authored or reviewed drafts of the paper, approved the final draft.

### Data Availability

Sequence data is available in the SRA database: PRJNA385181.

### Supplemental Information

Supplemental information for this article can be found online at http://dx.doi.org/10.7717/peerj.6609#supplemental-information.

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
