# Peer review of "Phytobiomes are compositionally nested from the ground up"

_PeerJ, doi:10.7717/peerj.6609_

## Round 0.1 · original submission · Major Revisions

Dear Anthony,

Your manuscript has been reviewed by two reviewers, who found that your work has merit but also suffers from some deficiencies. Upon careful consideration, I am ready to consider evaluating a revised version that would take into account seriously the comments and suggestions made.
In particular, please carefully address the following:

(1) Please acknowledge that this study only includes swabbing the outer plant tissues, and that this is thus reflective of microbial deposition and to a lesser degree of, actual differences in the microbiome tightly associated to plant compartments.
(2) More generally, please discuss whether sampling by swabbing and/or the low sequence number per sample may not provide adequate or enough data to differentiate communities between compartments. Rarefaction curves should be provided in this context.
(3) More information should be provided regarding the 16S rRNA gene library preparation (detail the PCR reaction conditions, the MiSeq sequencing procedures, and the process of rough sequences).
(4) Additional references should be cited (see suggestions by both reviewers).

I look forward to receiving a revised version of your manuscript along with a point by point response to the reviewers' comments and suggestions.

Best regards,
Xavier

Reviewer 1 ·

Basic reporting

I commend the authors for their clear and professional English used throughout this article. There are relevant literature cited throughout, although several sections, particularly in the introduction, could include additional citations to support background information given. Specifically, Line 65 (and throughout paragraph) states that other work does has yet to determine processes responsible for microbial assembly of plants, yet there are many publications that the authors cite in the beginning of the introduction that are doing just this; please provide relevant citations. See also Line 72, Line 85-91 (no citations are given).

Raw sequence data is shared through NCBI SRA database. This article is self-contained with relevant community ecological results regarding S. taccada microbiomes. However, their methodology of sampling provides somewhat limited inference on the microbiome, as only swabs were taken, even for soil samples, which may purely be from microbial propagule deposition non reflective of the actual endophytic microbiome (this latter approach would require tissue collection and laboratory sterilization). See additional details regarding these issues, among others, in the next section.

Experimental design

The research is within the Aims and Scope of PeerJ. Research questions were well-defined in the Introduction and certainly meaningful in terms of better understanding how microbial communities assemble in a tropical, native plant species. However, there are significant flaws not only in experimental design and sampling, but also in describing molecular methodologies used to prepare and analyze sequence libraries as described below:

1. Plant microbiomes, particularly with high-throughput applications readily available now (at least more so than in the recent past), in light of understanding below- and aboveground communities should be investigated by surface-sterilizing plant compartments (roots - stems - leaves, etc.; in other words, microbes that lie within tissue) and targeting gDNA within tissue. This study only includes swabbing the outer tissue which is going to be potentially primarily reflective of microbial deposition and to a lesser degree, actual differences in the microbiome among compartments. This seems to be demonstrated in the MDS with somewhat large overlap between some compartments that typically are much more differentiated (leaf versus rhizosphere/soil). This may reflective of (i) actual microbiome differences, (ii) an artifact of sampling by swabbing or (iii) low sequence yields (1000 seqs/sample) not providing enough data to differentiate communities.

2. There is relatively no information regarding 16S rRNA gene library preparation and no citations to accompany this section. More information regarding PCR reaction conditions are needed (or cite relevant papers which use such conditions) as well as MiSeq sequencing procedures (PhiX spike?). Since the authors used 2x300 cycles, they should have recovered better yields per sample if prepared adequately.

3. It is not clear how certain bioinformatic steps were completed. Line 162-163 states average quality scores declined after 150 bp and then "not further considered." Does this refer to trimming sequences, and if so, how did the authors do this (e.g., via QIIME, cut adapt, some other program)? Using a V3 kit, sequence quality shouldn't drop off at 150 bp one would think. Furthermore, did the authors investigate other contaminants besides chloroplasts (i.e., mitochondria, unclassified at Domain level) and did they include Archaea? What reference databases were used for alignment and taxonomy? If qiime default parameters were used, it is assumed greengenes, but this needs to be stated clearly.

4. Lastly, 1,000 sequence depth is very low for below ground samples. Assuming there is better sequence yield for more diverse and high biomass samples (rhizosphere, soils), perhaps the authors can, in addition to comparing all samples, separate out compartments (below ground vs aboveground) to achieve better yields and have a more accurate picture of soil communities. 1,000 sequences is not enough to describe vast diversity found in soils and may not be enough for aboveground communities as well. A rarefaction curve of all plant parts can answer this latter question.

Validity of the findings

Due to issues with methodologies as provided currently, I am not providing any comments regarding findings as I am uncertain whether they may be an accurate reflection of S. taccada microbial communities.

Reviewer 2 ·

Basic reporting

OVERVIEW

This manuscript examines how microbes are distributed within and between plants. Focusing on the beach shrub Scaevola taccada on the island of Oʻahu, it finds that geographic distance accounts for little variation in the composition of plant-associated microbial communities, but that part of the plant does (e.g., rhizosphere versus flowers). More specifically, it finds that microbial communities in plants are nested from the ground up, with microbial communities in flowers being subsets of microbial communities in the rhizosphere. A model of sequential colonization is proposed to account for this nested pattern.


COMMENTS

I enjoyed reading this paper and look forward to seeing it published. It is clearly written, and addresses a pertinent question using an interesting data set. It has only one substantial shortcoming, which should be relatively easy to address, that the citation of existing literature on nestedness is inadequate. In a general ecological context, there is an extensive literature on how to measure nestedness and on its interpretation. Much more of this literature should be cited both in the methods (e.g., the recent papers on partitioning beta-diversity into nestedness and turnover components), and in the discussion in interpreting the causes of the patterns that are reported (e.g., ordered extinctions as an alternative to ordered colonizations). Furthermore, in a microbial context, Thompson et al (Nature, 2017, doi:10.1038/nature24621) recently reported that on a global scale, plant microbiomes tend to be nested -- for instance, as shown here, it reported that plant-associated communities tend to be nested within rhizosphere communities. That said, although some results are similar, this work clearly adds to the results reported in Thompson et al (2017) and hence, the present manuscript just needs to discuss its findings in the context of the previous findings. Additional literature on the nestedness of microbiomes also exists (e.g., Smith et al, 2018, https://doi.org/10.1038/s41396-018-0086-0) and should also be discussed.

Experimental design

no comment

Validity of the findings

no comment

---

## Round 0.2 · accepted · Accept

Dear Anthony

Reviewers 1 and 2 have re-evaluated your ms and recommended Acceptance. I am happy with the revised version. In particular, I think it is good that you now make clearer that you've analyzed surfaces of plant parts.

Best regards
Xavier

# Reviewer 1 ·

Basic reporting

no comment

Experimental design

no comment

Validity of the findings

no comment

Reviewer 2 ·

Basic reporting

Pass

Experimental design

Pass

Validity of the findings

Pass

Additional comments

My concerns regarding the initial version of this manuscript have been adequately addressed and I will be happy to see it published.